# Earthquake-enhanced dissolved carbon cycles in ultra-deep ocean sediments

Mengfan Chu [1], Rui Bao [1] ✉, Michael Strasser [2], Ken Ikehara [3], Jez Everest [4], Lena Maeda[5], Katharina Hochmuth [6,7], Li Xu [8], Ann McNichol[9], Piero Bellanova [10], Troy Rasbury [11], Martin Kölling [12], Natascha Riedinger [13], Joel Johnson [14], Min Luo [15], Christian März[16,17], Susanne Straub [18], Kana Jitsuno[19], Morgane Brunet[20], Zhirong Cai[21], Antonio Cattaneo [22], Kanhsi Hsiung[23], Takashi Ishizawa[24], Takuya Itaki[3], Toshiya Kanamatsu[25], Myra Keep [26], Arata Kioka [27], Cecilia McHugh[28], Aaron Micallef[29], Dhananjai Pandey [30], Jean Noël Proust[20], Yasufumi Satoguchi [31], Derek Sawyer [32], Chloé Seibert[33], Maxwell Silver [34], Joonas Virtasalo [35], Yonghong Wang[36], Ting-Wei Wu [12,37] & Sarah Zellers [38]

Hadal trenches are unique geological and ecological systems located along subduction zones. Earthquake-triggered turbidites act as efficient transport pathways of organic carbon (OC), yet remineralization and transformation of OC in these systems are not comprehensively understood. Here we measure concentrations and stable- and radiocarbon isotope signatures of dissolved organic and inorganic carbon (DOC, DIC) in the subsurface sediment interstitial water along the Japan Trench axis collected during the IODP Expedition 386. We find accumulation and aging of DOC and DIC in the subsurface sediments, which we interpret as enhanced production of labile dissolved carbon owing to earthquake-triggered turbidites, which supports intensive microbial methanogenesis in the trench sediments. The residual dissolved carbon accumulates in deep subsurface sediments and may continue to fuel the deep biosphere. Tectonic events can therefore enhance carbon accumulation and stimulate carbon transformation in plate convergent trench systems, which may accelerate carbon export into the subduction zones.

Hadal trenches form due to downward bending of subducting ocean crust along subduction zones and represent the least-known ultra-deep oceanic environment[1]. Earthquakes, the frequent tectonic events at convergent plate boundaries, trigger redistribution of sediments that contain organic carbon (OC)[2–7] from the continental and trench slopes[8] into the trench axis. Sediments are efficiently transported into and focused at the trenches due to their V-shaped morphology, resulting in rapid accumulation of sedimentary organic carbon (SOC) in the trenches that is ~70 times higher than the global deep sea average[2,6,9].

Active OC cycling in hadal trench sediments may be triggered by the high-volume material inputs and intensive microbial activities[10–12].

SOC is remineralized to dissolved organic and inorganic carbon (DOC, DIC) in the sediment interstitial water (IW). In addition, the sulfate-methane transition zone (SMTZ) constrains the preservation and production of biogenic methane[13–15], serving as an OC diagenetic front in marine sediments[16]. The production and consumption of dissolved carbon in the hadal trench sediments are key processes in the deep subseafloor carbon cycle, supporting deep biosphere metabolisms. As carbon in trench sediments eventually enters the subduction zone system during plate convergence[1], the transformations of SOC, DOC, DIC, and methane may influence the carbon exchange among the Earth's surface biosphere, the lithosphere, and even the mantle[17].

In the Japan Trench, pioneering studies documented SOC delivery to the trench by historical subduction zone mega-earthquakes[2,6,18] with various magnitudes and recurrences on centennial-millennial time scales[4], however, their impacts on the cycling of the dissolved carbon in trench sediments remain largely unexplored. Here we present the IW DOC and DIC concentrations, carbon isotopic profiles ($\delta^{13}$C and $^{14}$C ages) and headspace methane concentrations in combination of typical IW geochemical parameters from 12 sites along the Japan Trench axis (water depth >7.5 km, Fig. 1a) collected during the International Ocean Discovery Program (IODP) Expedition 386[19]. Using this deep subsurface dataset from a hadal trench, we attempt to elucidate the accumulation and transformation of dissolved carbon in the hadal trenches, and to decipher the roles of tectonic events in the carbon cycles of these ultra-deep-water environments connected to subduction zones.

## Results and discussion

### Labile dissolved carbon accumulation in trench sediments

The Japan Trench receives pelagic background SOC descended from the water column and earthquake-remobilized SOC with various provenances (land, continental shelf, slope, and trench slope, etc.) and isotopic signatures[2,8]. Previous studies found the pervasive existence of earthquake-generated turbidite deposits along the Japan Trench[3,6,8]. At Sites M0081, M0083, and M0084, the $^{14}$C and $^{13}$C discrepancies between different carbon pools (Supplementary Fig. 1) exclude the possibility that the majority of the dissolved carbon in Japan Trench subsurface sediments are sourced from the exchange with bottom water or remineralization of pelagic background SOC. Instead, the dissolved carbon is mainly produced from remineralization of earthquake-introduced, likely marine-sourced SOC with various $^{14}$C ages and chemical recalcitrance (Supplementary Text 1). Additionally,

the high IW alkalinities along the trench axis (Fig. 1b) suggest that earthquake-triggered SOC accumulation[2] is followed by intensive SOC remineralization. The surficial (above 0.35 meter below the seafloor, mbsf) IW DOC concentrations (Fig. 2) in the Japan Trench are ~6 times higher than those in other oceanic trenches[20] and increase with depth, reaching maximum concentrations (Fig. 2) that are comparable to the coastal[21] and continental shelf[22] areas, highlighting a dynamic carbon cycle and huge dissolved carbon storage in the hadal trenches enhanced by earthquakes.

In many marine environments, most of the labile OC is selectively remineralized to DIC while the refractory fraction remains within the DOC pool[20,23,24], potentially leading to older DO$^{14}$C ages than DI$^{14}$C ages[23,25,26]. However, our data in the Japan Trench exhibit similar DO$^{14}$C and DI$^{14}$C ages (Fig. 3) that increase almost linearly with depth (Fig. 2). Essentially, this suggests a relatively fast equilibration of dissolved carbon pools supported by rapid IW diffusion[27] before their burial into the deeper subsurface. Additionally, no significant $^{14}$C discrepancy between the DOC and DIC pools in the hadal sediments means that at least part of the labile DOC is preserved, and go through aging in the trench sediments rather than being intensively oxidized or diffused overwhelmingly into the overlying seawater, in sharp contrast to other oceanic settings where mostly refractory DOC is preserved[23,24,28]. While the labile DOC in shallow subsurface sediments typically acts as a carbon source to the atmosphere due to microbial carbon turnover, those in the deeper-subsurface anoxic sediments tend to be chemically recalcitrant and elude aerobic oxidation[29]. In the Japan Trench, however, the instantaneous deposition of up to several meter thick event layers[4,30] driven by earthquakes prevents the exposure to oxidation and exchange with ocean water for most of the sediment[9,12], resulting in a chemically-active dissolved carbon sink in trench sediments.

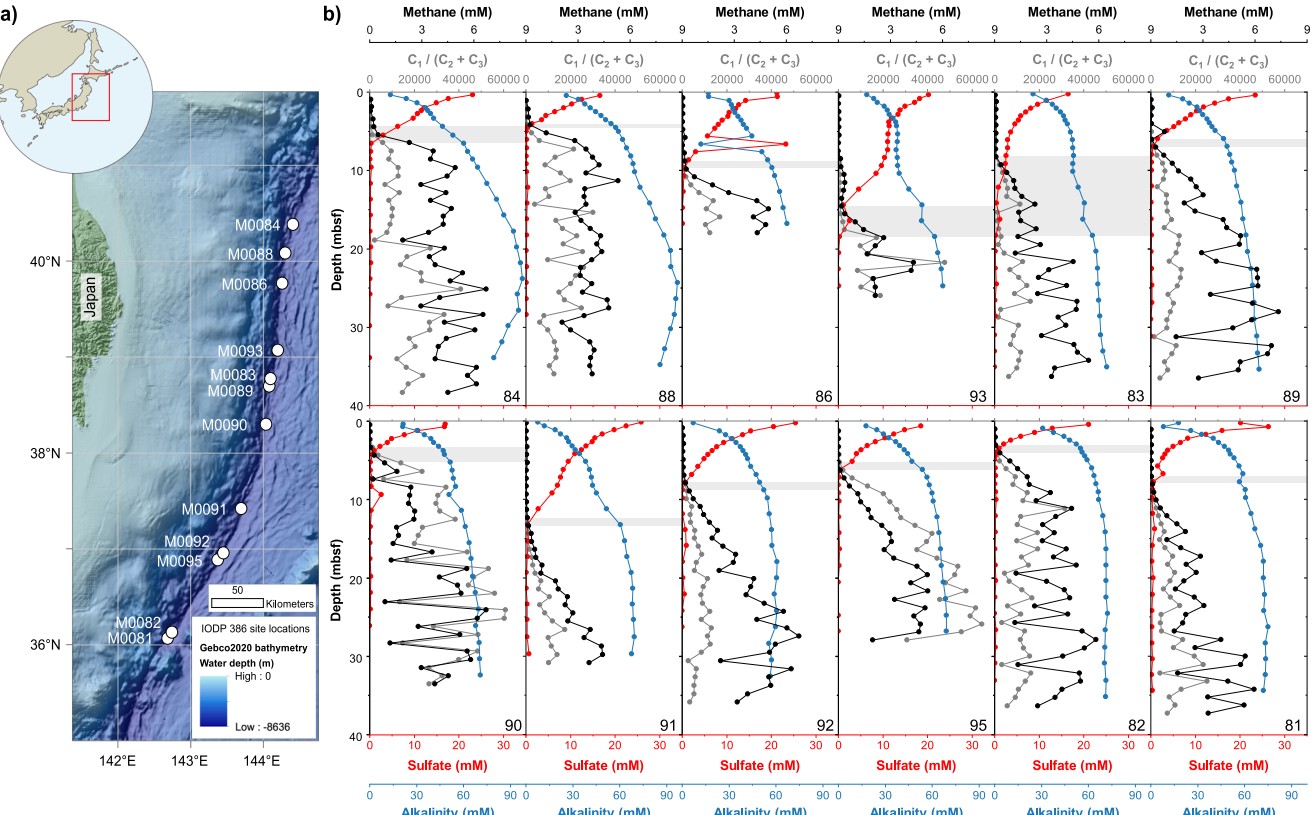

**Fig. 1 | Site map in the Japan Trench and IW geochemical parameters. a** Location of twelve IODP 386 sites; **b** Alkalinity (blue), sulfate (red) and methane concentrations (black), and the ratios of methane to ethane and propane ($C_1/(C_2 + C_3)$,

gray) at the studied sites. The SMTZ are inferred by depths with minimum sulfate and methane concentrations and marked in gray. Source data are provided as a Source Data file.

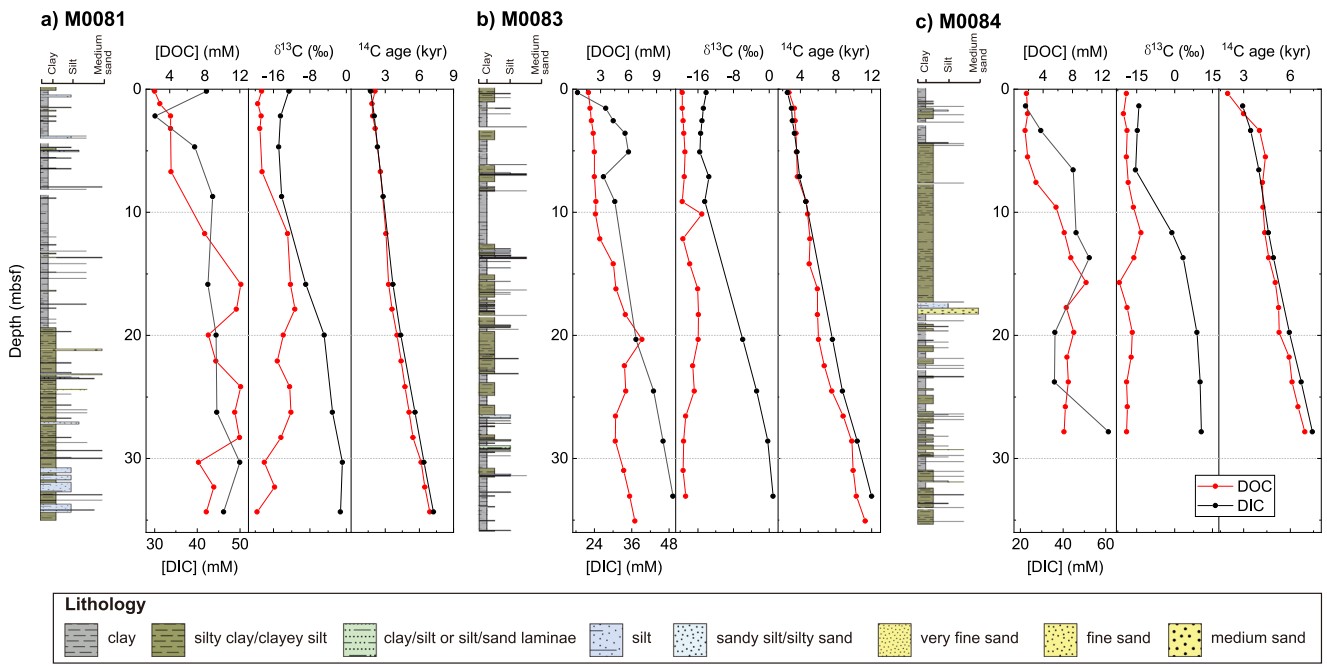

**Fig. 2 | Lithologies, IW DOC (red) and DIC (black) concentrations, δ¹³C values and ¹⁴C ages of sediment cores at three IODP 386 Sites.** Sites **a** M0081, **b** M0083 and **c** M0084 are collected from the southern, central and northern Japan Trench, respectively. Source data are provided as a Source Data file.

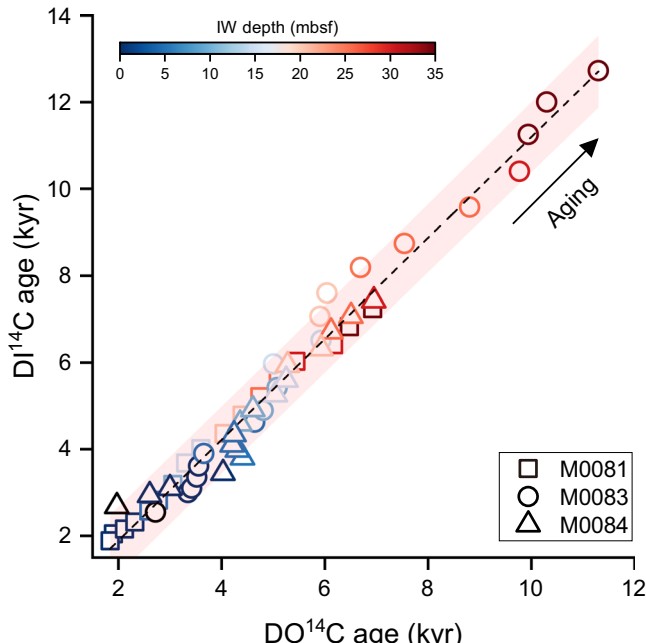

**Fig. 3 | DI¹⁴C ages vs. DO¹⁴C ages of IW samples from Sites M0081 (square), M0083 (circle), and M0084 (triangle).** ¹⁴C ages exhibit a linear relationship between DOC and DIC ($R^2 = 0.98$, $p < 0.01$). The pink shading illustrates 95% confidence band of the linear regression. IW depth is indicated by color bar. Source data are provided as a Source Data file.

Although the IW DOC is derived from earthquake-induced SOC remineralization, our results show that the relatively constant production of DOC may not be completely regulated by SOC content (Supplementary Fig. 2). It is possible that only the most labile fraction of the SOC is remineralized into DOC, supported by the positive δ¹³C_DOC evidence (Fig. 2). Despite processes such as remineralization and methanogenesis that consume IW DOC, we still find sufficient residual DOC that accumulates and ages with depth (Fig. 2), which

remains available for biogeochemical processes in the deeper subsurface sediments. In addition, the linearly-increasing ¹⁴C ages of the residual DOC indicate the time since sediment deposition and may constrain the chronological models for trench sediments devoid of pelagic carbonates. This implies a steady state DOC turnover in a non-steady state oceanic sedimentary setting that is irregularly disturbed by turbidite deposition. The dissolved carbon pools in the trench sediments thus provide potential information about the hadal sedimentation and diagenesis processes.

### Enhanced methanogenesis in hadal zones

The large amount of dissolved carbon may fuel microbial metabolisms in the deep subsurface sediments[31,32]. Below the SMTZ, the methane concentrations along the Japan Trench (up to 7.35 mM, Fig. 1b) and the high ratios of methane to ethane and propane ($C_1/(C_2 + C_3) > 1000$, Fig. 1b) imply intensive microbial methanogenesis[33]. The abundant DIC in the deep subsurface sediments accumulated from intensive OC remineralization has the potential to promote methanogenesis through $CO_2$ reduction. In turn, methane production and oxidation exert significant control on the DIC pools below and above the SMTZ, respectively (Supplementary Fig. 3 and Supplementary Text 1). The positive δ¹³C_DIC values in the deep subsurface sediments (Fig. 2) indicate a carbon isotope fractionation during microbial methanogenesis, where the residual DIC becomes more ¹³C-enriched[33,34]. Up to 24%–38% of the DIC is estimated to be utilized during methane production in the Japan Trench (Supplementary Fig. 4, see Method for calculation), highlighting intensive methanogenesis processes fueled by the accumulated DIC in the trench sediments.

Low concentration and short residence time of dissolved substrates may result in relatively less significant methanogenesis through fermentation in marine environments[33,35]. However, below 30 mbsf, the DOC is younger than DIC in the Japan Trench (Fig. 3), likely suggesting that a newly-added DOC pool is more bioavailable to heterotrophic organisms[36]. The labile DOC accumulation implies the possibility of intensified methanogenesis through fermentation[37] in hadal trenches along subduction zones. The thick earthquake-triggered turbidite deposit results in less permeability[38] and prevents

oxygen and sulfate supplies to the sediments below it, leading to intensive OC remineralization[12] and elevated SMTZ, thereby enabling substrate availability for in-situ microbial fermentation. Enhanced sediment compaction by repeated deposition of thick turbidites[30] further contributes to these processes. The earthquake-triggered labile DOC production and preservation thus fodder microbial methanogenesis in the deep subsurface sediments.

With increasing time after deposition and repeated occurrences of large earthquakes, the IW volume is reduced due to sediment compaction and earthquake-induced dewatering and consolidation[39],

resulting in relatively constant DOC concentration below 20 mbsf (Fig. 2). The maximum concentration at 15~20 mbsf (Fig. 2) may indicate a compensation depth of the labile DOC, below which the DOC is consumed faster than it is produced and condensed, and may be continuously converted to methane by the microbial communities in deeper subsurface sediments[40]. The cascading effects of earthquake-induced processes including voluminous SOC deposition, rapid sediment burial, and sediment compaction consequently stimulate microbial methanogenesis via multiple metabolic pathways in the trench sediments (Fig. 4a).

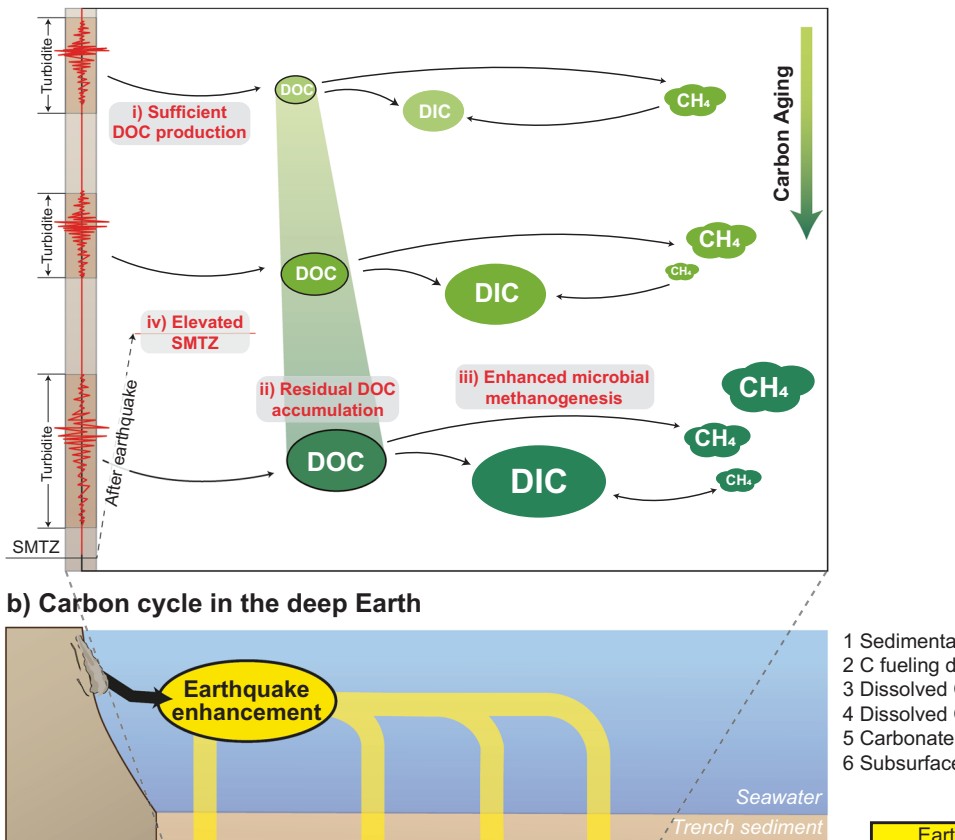

**Fig. 4 | Conceptual models of the earthquake-enhanced carbon cycle in trench sediments and the shallow subduction zone. a** Dissolved carbon dynamics in trench sediments are enhanced by earthquakes. (i) Sufficient dissolved carbon production facilitated by the combined effects of earthquake-triggered sediment deposition and compaction leads to (ii) larger DOC and DIC pools that are aging with depth, which result in (iii) enhanced microbial methanogenesis via

fermentation and $CO_2$ reduction, and (iv) elevated SMTZ that enables methanogenesis at shallower depth. **b** The impact of earthquakes on the carbon cycle in the subduction zone. The purple star indicates megathrust earthquakes along the plate boundary, which trigger seismic remobilization of SOC to the trench. Carbon in the trench sediments enters the subduction zone and undergoes dehydration, forming carbon reservoirs during processes in the deep Earth.

## Linking between shallow and deep carbon cycles

The OC preservation regulated by the penetration of electron acceptors (oxygen, sulfate, etc.) is usually correlated with water depth in deep-sea environments[41]. The OC preservation and turnover in the hadal trenches, instead, are regulated by earthquake disturbances. The earthquake-triggered process cascade prolongs the residence time of labile carbon, which would have been otherwise consumed and released into the bottom water on the time scale of centuries to millennia, and thereby promotes carbon aging and subduction in the trench sediments (Fig. 4a).

The labile DOC pools serve as food sources for microorganisms in the trench sediments and have the potential to fuel the microbial-mediated carbon transformation in the deep biosphere[42,43] (Fig. 4b). Volcanic ashes, biosiliceous-rich sediments, detrital silicate minerals and the abundant alkalinity along the Japan Trench (Fig. 1b) are suggestive of a contribution from marine silicate weathering, which provides the alkalinity and cations needed for authigenic carbonate precipitation[44] (Supplementary Fig. 5). This deeper-subsurface carbonate formation serves as a 'carbon sink swap' from OC to methane, carbonates and eventually to graphite as more stable forms in the subduction zone system[45,46]. Although substantial storage of methane has been reported in the continental shelf and cold seep sediments[47], its presence in trench sediments is not thoroughly evaluated. A comparison of the methane data from the IODP database (Supplementary Fig. 6) suggests that the methane storage in the Japan Trench may be comparable to typical methane-rich seafloor areas. More methane supply from the deep subsurface would support stronger methane oxidation in the SMTZ and reduce the sulfate available for OC remineralization, thereby promoting deep carbon burial.

The earthquake-enhanced dissolved carbon storage in the trench sediments may eventually contribute to carbon subduction[1,48]. Through sediment deposition and compaction, earthquake-triggered turbidites instantaneously increase the depth of the carbon residing in the sediments, leading to intensified carbon cycling in deeper subsurface depth than expected. The dissolved and sedimentary carbon may enter the subduction zones without becoming chemically recalcitrant for utilization by the deep biosphere[12]. The pore fluid containing dissolved carbon enters the crustal and mantle parts of the subducting slab and is transformed into solid phases, forming various carbon reservoirs (carbonates[49], sediment-derived fluid[50], graphite[46,51] etc., Fig. 4b) at forearc depth[52,53]. Earthquakes therefore accelerate and magnify the processes in the subduction zones (Fig. 4b). In the meantime, spatial heterogeneity of our IW geochemical results exists along the trench. Considering the global variations of sedimentation environments, SOC characteristics, and earthquake frequencies in the trench systems[11,54,55], we suggest that the spatial and temporal dynamics of dissolved carbon reservoirs will be further amplified by irregularity of earthquake occurrence and difference in their magnitudes in trenches globally, and will therefore cause critical and variable impacts on the global carbon cycle.

Our presented data and results, acquired through advanced drilling and radiocarbon measurement techniques, offer a comprehensive perspective of the hadal trench carbon cycle in the deepest and least explored environments on our planet, where geological events such as great subduction zone earthquakes may serve as key modulators. The contribution from our newly-discovered carbon reservoirs in the Japan Trench sediments provides essential, detailed information regarding how intensive earthquake-induced SOC accumulation leads to enhanced microbial activities in the deep biosphere. The occurrence of the dissolved carbon compensation depth indicates intensive carbon maturation and alteration in the trench subsurface sediments. These processes occur at the inlet to the subduction zone systems, where fresh materials are irregularly introduced by geological events. The cascade of the earthquake-triggered physical and chemical processes accelerates the transformation between carbon in different forms (i.e., 'carbon sink swap') at the shallowest setting of the subduction zone

systems. Meanwhile, open questions remain regarding whether, and how much of, the carbon in the trench sediments is subducted in its various forms. We suggest that the earthquake-enhanced dissolved carbon reservoir in the trenches will constitute a potential source for the deep carbon reservoir in the lithosphere or mantle.

## Methods

### Research materials

The Japan Trench is located between the Kuril-Kamchatka Trench and the Izu-Bonin Trench. It is formed by the subduction of the Pacific Plate beneath the Okhotsk Plate with a convergence rate of 8.0 – 8.6 cm/yr (ref. 56). It is 611 km long and has a surface area of 37,854 km$^2$ (ref. 57). Normal fault grabens resulting from the flexural bending of the slightly obliquely subducting Pacific Plate oceanic crust form numerous isolated basins that act as terminal sinks for trench-fill sedimentation along the axis of the Japan Trench[30,58]. Sediment cores were collected with up to 40 m long giant piston corers (GPC) on R/V *Kaimei* during the International Ocean Discovery Program (IODP) expedition 386 in 2021 (Fig. 1a & Supplementary Table 1).

### Sample collection

Interstitial water (IW) was collected on board R/V *Kaimei* immediately on recovery using Rhizone samplers (CSS 19.21.23 F; Rhizosphere Research Products, Netherlands). Rhizone samplers consisted of the microporous tube (2.5 mm diameter, 5 cm length) that was supported by a nylon wire/PEEK wire and connected to PVC tubing with a female Luer lock connector, that can be attached to a syringe. Before use, Rhizone samplers were soaked in purified lab water for approximately 60 minutes. A 3.8 mm hole was drilled into the plastic core liner using the drill bit, and the Rhizone filament was inserted into the sediments. Negative pressure was applied by attaching a 24 mL plastic pulled-back syringe to the Rhizone sampler. The sediment cores remained capped throughout this process to minimize ambient gas penetration. First 0.1 mL (1 – 2 drops) were discarded because it would include purified lab water, and then Rhizone samplers were reassembled and remained in the cores for a maximum 6 h. The IW was split into sub-samples for alkalinity (3–4 mL) and sulfate (3 mL) measurements following IODP standards[19], and for dissolved organic carbon (DOC, 8 mL) and inorganic carbon (DIC, 2 mL) measurements. Split samples for alkalinity and sulfate measurements were stored at 4°C. Sample splits were added with 50 μL 10% HCl and stored at −20°C for DOC measurement, or were added with 10 μL saturated HgCl$_2$ and stored at 4°C for DIC measurement.

Bottom water (BW) was collected from the top of the pilot cores using a BW sampling system consisting of a Duran bottle and vacuum pump unit connected by tubes. Negative pressure was applied and BW samples were sucked into the Duran bottle through the tube placed in the overlying water of the trigger core. After filtering using a Rhizon sampler, BW samples were split for DOC (1 L) and DIC (5 mL) measurements. Sample splits were added with 1 mL 10% HCl and stored at −20°C for DOC measurement, or were added with 10 μL saturated HgCl$_2$ and stored at 4°C for DIC measurement.

Headspace samples were taken directly from the base of each core section after core cutting. A total volume of 5 cc (using 2.5 cc cut tip syringe twice) was collected and each 2.5 cc sample was then added into 20 mL crimp vial with 5 ml of 1 M NaOH and stored upside down and cooled at 4°C.

### Geochemical analysis

IW and BW geochemical analysis was carried out during the onshore and offshore phases of the IODP Expedition 386 following standard IODP procedure[19].

Water splits were analyzed onboard for alkalinity immediately following extraction with an autotitrator (Metrohm 888 Titrando). An aliquot of 1–3 mL water split-adjusted to 3 mL with 0.7 M KCl solution

was titrated with nominal 0.1 M HCl at 25 °C. A 100 mM $Na_2CO_3$ solution was used for the calibration of the acid each week. Quality checks were conducted using a 50 mM solution of $NaHCO_3$ and 0.1 M HCl once per day.

Water splits were measured for sulfate on a Metrohm 882 Compact IC ion chromatograph at the MARUM. A 40-fold dilution of IAPSO seawater and standards prepared from commercial single anion standards was used for calibration. Analytical precisions were ± 0.95% for sulfate.

Gas concentrations in headspace samples were measured between September and November 2021 onboard of D/V *Chikyu*. The samples were vigorously shaken by hand for 2 minutes, then shaken by a lab shaker for 1 h and left for 23 h at room temperature. Each vial was placed in an Agilent 7697 A headspace sampler, and was heated to 70 °C for 30 minutes. An aliquot of the headspace gas was then automatically injected (split mode) into an Agilent 7890B gas chromatography equipped with a packed column (HP PLOT-Q) and linked to a flame ionization detector. The carrier gas helium flow rate was 10 $cm^3$/min. The oven temperature was programmed to start at 60 °C and was ramped to 150 °C at a rate of 10 °C/min. The instrument was calibrated for the chromatographic response with commercial standards (Types VIII–XII, GL Sciences, Japan). Quantification of hydrocarbon gases was achieved by comparison of its chromatographic response with a three-point calibration curve.

## DOC and DIC concentration and isotopic analysis

Holes M0081D, M0083D, and M0084D were chosen to represent the DOC and DIC signatures of the corresponding sites (Supplementary Table 1). All DOC and DIC samples were analyzed at the National Ocean Science Accelerator Mass Spectrometer (NOSAMS) facility at Woods Hole Oceanographic Institution for concentration, $^{13}C$ and $^{14}C$ analyses. Due to the pandemic, water splits were stored at the Kochi Institute for Core Sample Research of the Japan Agency for Marine-Earth Science and Technology for 6 months before being shipped to NOSAMS. The DIC concentration was likely to be affected by some carbonate precipitation during storage given the extreme IW alkalinities of up to 100 mM. The DOC and DIC measurement procedures were described in detail in refs. [59],[60]. Briefly, BW samples were filtered through precombusted 0.45 μm GF/F films before measurement. A quartz reactor prefilled with Milli-Q water was irradiated with UV for 1.5 h to reduce the potential contribution of extraneous carbon. Approximately 2 mL water sample and 0.4 g 33% ultra-pure hydrochloric acid were then added to the reactor. DOC samples were irradiated with UV for 3 hr. The generated $CO_2$ was purged by ultra-high purity helium (UHP He) at 120 mL/min for 66 min, and was cryogenically trapped, purified, manometrically quantified, and collected by flame sealing in a 6.4 mm OD Pyrex glass tube. DIC was also extracted on the DOC line by replacing the quartz reactor with a gas-washing device equipped with an injection port. The device was added with Milli-Q water and $H_3PO_4$ and then cleaned with He before sample injection. The generated $CO_2$ was purged by UHP He at 120 mL/min and collected by a $CO_2$ trap.

$\delta^{13}C$ were measured by a VG Optima or VG prism Isotope Ratio Mass Spectrometer at NOSAMS. $\delta^{13}C$ values were reported in ‰ relative to the VPDB standard. The typical precision of $\delta^{13}C$ measurement was between 0.1‰ – 0.23‰. $^{14}C$ were measured by a continuous-flow accelerator mass spectrometer (CFAMS) system constructed around an accelerator (National Electrostatics Corporation, Middleton, WI, model 1.5SDH-1) at NOSAMS[61],[62]. AMS uncertainty was calculated as the larger of either the statistical uncertainty using the total number of $^{14}C$ counts measured for the target, or an uncertainty calculated from the reproducibility of multiple measurements of the target, both propagated with uncertainties from the normalizing standards and blank subtraction. $\Delta^{14}C$ values were reported as $^{14}C$ ages[63]. For the IW samples, the $^{14}C$ measurement errors were within 0.34% – 2.03% and thus didn't impact data robustness.

## DIC conversion estimation

During methanogenesis via $CO_2$ reduction in the marine environment, DIC is utilized by methanogens with a kinetic isotope effect that discriminates $^{13}C$. The fractionation factor of DIC to methane ($\varepsilon_c$) is between −50 to −90‰ in natural observations[33]. Due to the lack of $\delta^{13}C_{methane}$ data, we use both reported $\varepsilon_c$ values[33] as upper and lower limits to estimate the amount of DIC that is converted to methane. A Rayleigh distillation function can be used[40] to describe the $^{13}C$ fractionation during methanogenesis in sediment IW:

$$\delta^{13}C_{DIC,t} - \delta^{13}C_{DIC,0} = -\varepsilon_c \ln f_{residual} \qquad (1)$$

Where $\delta^{13}C_{DIC,0}$ and $\delta^{13}C_{DIC,t}$ are the isotopic compositions of the initial and residual DIC pools at a certain time, respectively, and $f_{residual}$ represents the amount of residual DIC expressed as a fraction ($f$ ranges from 0 to 1). The fraction of DIC that has been utilized during methanogenesis, $f_{converted}$, can be calculated as:

$$f_{converted} = 1 - f_{residual} \qquad (2)$$

We use the linearly-extrapolated $\delta^{13}C_{DIC}$ at 0 mbsf (−12.43‰, −14.00‰ and −13.58‰ for Sites M0081, M0083 and M0084, respectively) as $\delta^{13}C_{DIC,0}$. The fraction of residual DIC can thus be calculated according to Eq. (1) by using the $\delta^{13}C_{DIC}$ at the given depth as illustrated in Supplementary Fig. 4. Calculation results of the deepest IW sample of each Hole are presented in Supplementary Table 2.

## Reporting summary

Further information on research design is available in the Nature Portfolio Reporting Summary linked to this article.

## Data availability

Data used in this study are presented in the Supplementary Information/Source Data file, and are also available on Figshare https://doi.org/10.6084/m9.figshare.23929950.v2. Source data are provided with this paper.

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

## Acknowledgements

This research used samples and data provided by the International Ocean Discovery Program (IODP) that are under moratorium until 30 November 2023. After the moratorium period, further data will be accessible online and more samples will be made available on request (www.iodp.org/access-data-and-samples). We thank staffs for assisting with sample preparation and measurement, and Nan Wang for comments on the figures. Expedition 386: Japan Trench Paleoseismology was jointly funded by the European Consortium for Ocean Research Drilling (ECORD), the U.S. Science Support Program, and the IODP, with contributions and logistical support from the Japan Agency for Marine-Earth Science and Technology (JAMSTEC). Any opinions, findings, conclusions, or recommendations expressed in this publication are those of the author(s) and do not necessarily reflect the views of the funders. This research is funded by National Natural Science Foundation of China (92058207, 42076037 and 42242601, R.B.), Fundamental Research Funds for the Central Universities (202042010, R.B.), the National Key Programs for Fundamental Research and Development (2016YFA0600904, R.B.), Taishan Young Scholars (tsqn202103030, R.B.), and Shandong Natural Science Foundation (ZR2021JQ12, R.B.).

## Author contributions

R.B. designed this work. M.C. and R.B. drafted the figures and wrote the original draft. M. Strasser and K.I. led the IODP 386 expedition. Administration contributions are from J.E., L.M., and K. Hochmuth. R.B., and L.X. conducted the isotopic measurement of DOC and DIC. All authors including M.C., R.B., M. Strasser, K.I., J.E., L.M., K. Hochmuth, L.X., A. McNichol, P.B., T.R., M. Kölling, N.R., J.J., M.L., C. März, S.S., K.J., M.B., Z.C., A.C., K. Hsiung, T. Ishizawa, T. Itaki, T.K., M. Keep, A.K., C. McHugh, A. Micallef, D.P., J.N.P., Y.S., D.S., C.S., M. Silver, J.V., Y.W., T.W., and S.Z. contributed to the interpretation of data and the editing of the manuscript. All members of the IODP 386 Science Party contributed to sample acquirement, geochemical analysis and data collection.

## Competing interests

The authors declare no competing interests.

## Additional information

[1]Frontiers Science Center for Deep Ocean Multispheres and Earth System, and Key Laboratory of Marine Chemistry Theory and Technology, Ministry of Education, Ocean University of China, Qingdao 266100, China. [2]University of Innsbruck, Institute of Geology, Innsbruck, Austria. [3]National Institute of Advanced Industrial Science and Technology (AIST), Geological Survey of Japan, Institute of Geology and Geoinformation, Ibaraki 305-8567, Japan. [4]British Geological Survey, Lyell Centre, Edinburgh EH14 4AP, UK. [5]Center for Deep Earth Exploration, Japan Agency for Marine-Earth Science and Technology, Kanagawa 236-0001, Japan. [6]School of Geography, Geology and the Environment, University of Leicester, Leicester, UK. [7]Australian Centre for Excellence in Antarctic Sciences, Institute for Marine and Antarctic Studies, University of Tasmania, 20 Castray Esplanade, Battery Point TAS, Churchill Ave 7004, Australia. [8]NOSAMS Laboratory, Woods Hole Oceanographic Institution, Massachusetts, USA. [9]Department of Geology and Geophysics, Woods Hole Oceanographic Institution, Massachusetts, USA. [10]RWTH Aachen University, Institute of Neotectonics and Natural Hazards & Institute of Geology and Geochemistry of Petroleum and Coal, 52056 Aachen, Germany. [11]Stony Brook University, Department of Geosciences, New York 11794, USA. [12]MARUM – Center for Marine Environmental Science, University of Bremen, Bremen 28359, Germany. [13]Boone Pickens School of Geology, Oklahoma State University, Oklahoma 74078, USA. [14]University of New Hampshire, Department of Earth Sciences, New Hampshire 03824, USA. [15]Shanghai Engineering Research Center of Hadal Science and Technology, College of Marine Sciences, Shanghai Ocean University, Shanghai, China. [16]School of Earth and Environment, University of Leeds, Leeds LS2

9JT, UK. [17]Institute for Geosciences, University of Bonn, Nussallee 8, 53115 Bonn, Germany. [18]Lamont Doherty Earth Observatory, Geochemistry Division, New York 10964, USA. [19]Department of Life Science and Medical Bioscience, Waseda University, Tokyo 162–0041, Japan. [20]Univ Rennes, CNRS, Géosciences Rennes, UMR 6118, 35000 Rennes, France. [21]Kyoto University, Department of Geology and Mineralogy, Division of Earth and Planetary Sciences, Graduate School of Science, Kyoto 606-8502, Japan. [22]Geo-Ocean, UMR 6538, Univ Brest, CNRS, Ifremer, Plouzané F-29280, France. [23]Research Institute for Marine Geodynamics, JAMSTEC, Marine Geology and Geophysics Research Group, Subduction Dynamics Research Center, Kanagawa 237-0061, Japan. [24]International Research Institute of Disaster Science, Tohoku University, Sendai 980-0845, Japan. [25]Japan Agency for Marine-Earth Science and Technology (JAMSTEC), Research Institute of Marine Geodynamics (IMG), Yokosuka 237–0061, Japan. [26]The University of Western Australia, Department School of Earth Sciences, Perth, Australia. [27]Kyushu University, Department of Earth Resources Engineering, Fukuoka 819–0395, Japan. [28]Queens College, City University of New York, School of Earth and Environmental Sciences, New York 11367, USA. [29]GEOMAR Helmholtz Centre for Ocean Research Kiel, Kiel D-24148, Germany. [30]National Centre for Polar and Ocean Research, Ministry of Earth Sciences, Government of India, Goa 403 804, India. [31]Lake Biwa Museum, Shiga 525-0001, Japan. [32]The Ohio State University, School of Earth Sciences, Ohio 43210, USA. [33]Lamont Doherty Earth Observatory, Marine geology and geophysics division, New York 10964, USA. [34]Colorado School of Mines, Hydrologic Science and Engineering, Colorado 80227, USA. [35]Geological Survey of Finland (GTK), Espoo 02151, Finland. [36]Ocean University of China, Department of Marine Geosciences, Qingdao 266100, China. [37]Norwegian Geotechnical Institute, Oslo, Norway. [38]University of Central Missouri, Department of Physical Sciences, Missouri 64093, USA. ✉e-mail: baorui@ouc.edu.cn

