## [Peer Review File · Nature Communications]

REVIEWER COMMENTS

Reviewer #1 (Remarks to the Author):

This is a well-written and interesting manuscript providing new insights on the diagenesis and burial of organic carbon in the Japan trench. Hadal trenches are apparently unproportionally important for marine (and global?) element cycling due to; material focusing, seismic driven mass wasting and potentially translocation of material by tsunamis. But our current understanding on the processes governing transport, deposition and mineralization at these extreme depths are still rudimentary – but research interests are increasing. The study is timely, and the Japan trench is a very interesting case study for the biogeochemical function of hadal trench systems.

The data set is impressive, and the main conclusions are well substantiated. The finding complements other recent studies of hadal trench system, but also provide original and novel findings. Personally, I value the strong evidence of active early diagenesis (methanogenesis) in deep subsurface settings drive by relatively labile DOC. At times I feel the authors over sell some of their findings – or at least they should acknowledge other recent advances that also demonstrate i) intensified burial of Org C in hadal trench system and ii) subsurface burial of labile organics in hadal sediment fueling intensified microbial activity. I understand that the hadal research is advancing rapidly, and that recent publications may not find their way into manuscripts that are being developed concurrently. But fortunately, the authors still have a chance to do that.

In conclusion, recommend the manuscript is accepted with relatively minor revision. Most importantly the authors should place there study in the context of other recent and highly relevant publications.

More specific comments are listed below.

Abstract:

Line 79; I suggest changing “most mysterious” to “unique”.

Line 86-89 & 90-93; I acknowledge the novel contribution of this high-quality study (See above/below). But conceptually, the fact that earthquake-triggered turbidity in hadal trenches result in subsurface layers of labile organic material and intensified microbial activity – and intensified burial and subduction of organic material has been documented and discussed in recent publications. For example;

Zabel et al. (2022) High carbon mineralization rates in subseafloor Hadal sediments – result of frequent mass wasting. *Geochem, Geophys, Geosystem* 23. Where some of the evidence originates from gravity cores that have been recovered from the Japan trench and include the SMTZ. I am particularly surprised that this paper isn't considered in this submission.

Oguri et al (2022) Sediment accumulation and carbon burial in four hadal trench systems. *JGR Biogeosciences* 127. The manuscript, target more sites along different trench systems, it is estimated that carbon burial rates in hadal trenches are 70 times higher than the global average for the deep ocean.

Schauberger et al (2021) Spatial variability of prokaryotic and viral abundance in the Kermadec and Atacama trench regions. *Limnol Oceanogr* 66: 2095-2109.

None of these publications are acknowledged or cited in the current manuscript., but I find that the current submission should place their findings in the context of these recent advances.

Line 154-159; This point has been presented and discussed in two recent papers of Zabel et al 2022 & Oguri et al 2022. Referencing and relating to these studies would substantiate your claims.

Line 194-202; Can these claims be substantiated (no references are included, but they are presented as facts), or is these hypotheses? Please be specific.

Line 211-213; Evidence in the form of sequencing data (barcoding or metagenomics) or specific biomarkers for methanogens would have strengthened the proposed concept significantly. I don't know if there are any initiatives pursuing this for the recovered samples

Line 220; I suggest changing "atmosphere" to "bottom water". I acknowledge that "quickly" may mean different things for different people. But "quickly is unspecific and venting of deep oceanic waters is on time scales of centuries to millennia.

Line 227-231; Are there any hardcore evidence that authentic carbonate precipitation is occurring in hadal sediments? Or is this another hypothesis/claim? Please be specific.

Line 236 ; A reference here would be appropriate

Line 254-256; that is not correct – there have been more “coordinated international research efforts that perform high spatiotemporal resolution investigations and sampling of a hadal oceanic trench” For instance, the cruises SO261 and TAN1711 – that explored the deposition dynamic, carbon burial rates and carbon mineralization rates as well as microbial communities structures in sediment along the axes of the Atacama Trench and the Kermadec trench and relating condition to adjacent abyssal refences sites – a long list of papers from these expeditions have been published and more are in the pipeline. I fully acknowledge that this was not the impressively long cores that you obtain with the I ODP efforts but “only” gravity cores, multiples cores and box cores. But the spatiotemporal variability along the trench axes and linkage to mass wasting ws a core focus of these expeditions.

I suggest eliminating this sentence.

Review for Bao

I really appreciate to read a follow up study of the paper by Bao et al. On Hadal zone from a few years ago. It gives a sense of active progress on a key issue. That of turbidities and their role in transport (most?) organic carbon to trench.

The introduction is very clear, the goal is clearly stated.

I felt that the text is very interesting, but it reads sometimes like a review. You may spend a bit more time on your figures, explain very clearly what are your data, and do not mix so much your own work with that of your colleagues. This would bring a bit more clarity and focus to the main text.

I was fascinated by the process oriented study in hadal zone. The topic is very important, and the authors may be the only one focusing on this. The article reports a DOC compensation depth in the Japan hadal zone, they have evidences for carbon sink swaps (from organic carbon to carbonate) which is all very fundamental and important.

The issue I see is that some may say "we already know carbon matures and produces a bit of methane". And I would agree that the way the text is written does not really answer this potential criticism, or at least does not make clear enough why such criticism does not affect the work. What you did here is not just show the OC transforms and produces CH₄. You should highlight the combination of tools, the focus on a very hard to study field site, that you have actually advanced our understanding of OC maturation in this very specific location. You should answer those points right away, be clear about your contribution.

The conclusion is disappointing. It is full of "platitudes" and generalities unrelated to your work and not sexy at all (to me), they actually undermine your work greatly. What's sexy in your work is the novelties, the creative concepts blend with solid data, the amount of work you did. You should conclude your work with a big highlight on some of the key findings you made, and trust that we understand this is important. We do not need generalities to feel secure. Make us insecure instead, and show us that you are pushing forward in directions that no one is even looking at...

I think the study would benefit from a bit more work on the structure: your data first, and then a bit more discussion. Here you tried to merge everything to an extreme degree and you lost clarity in my opinion, it dilutes your contribution.

I'd be happy to check a final revised version.

"supporting the deep biosphere metabolites" does not really make sense. Metabolisms maybe.

L207: "compensatio deth of labile DOC" I find this notion intriguing.

L215 – 222: very interesting!

L227: do you mean authigenic?

L227-229: for a "carbon sink swap" the other way around, check out Galvez et al 2012: carbonate to graphite.

L260-269: very disappointed by this: this sounds like this could have been written elsewhere. I felt throughout that your research strength is that it focuses on a the detailed processes happening in the hadal zone, the inlet to the subduction system. Even though everything you mention is correct, I would not conclude this work by those generalities. Conclude by stressing again your contribution. Mention again the notion of a DOC compensation depth, the notion that carbon sink swap may exist as shallow as on the seafloor and pore water. Those notions and the data that you provide to support them are truly innovative, finish with that.

Response to Reviewers

We deeply appreciate the constructive comments from the Reviewers concerning our manuscript entitled “*Earthquake-enhanced dissolved carbon cycles in ultra-deep ocean sediments*”. These comments are very helpful for improving our manuscript. In this point-by-point *Response to Reviewers* we have addressed all the comments and suggestions. The responses to the Reviewers’ comments are listed below, with comments listed first in *blue italics*, followed by our response in black. We are pleased to provide the revised manuscript and hope both Reviewers are satisfied with our responses.

Comments from Reviewer #1

This is a well-written and interesting manuscript providing new insights on the diagenesis and burial of organic carbon in the Japan trench. Hadal trenches are apparently unproportionally important for marine (and global?) element cycling due to; material focusing, seismic driven mass wasting and potentially translocation of material by tsunamis. But our current understanding on the processes governing transport, deposition and mineralization at these extreme depths are still rudimentary – but research interests are increasing. The study is timely, and the Japan trench is a very interesting case study for the biogeochemical function of hadal trench systems.

Response: We are grateful for your efforts on reviewing our manuscript and precisely highlighting the context of our study.

The data set is impressive, and the main conclusions are well substantiated. The finding complements other recent studies of hadal trench system, but also provide original and novel findings. Personally, I value the strong evidence of active early diagenesis (methanogenesis) in deep subsurface settings drive by relatively labile DOC.

Response: Thank you for your positive comments.

At times I feel the authors over sell some of their findings – or at least they should acknowledge other recent advances that also demonstrate i) intensified burial of Org C in hadal trench system and ii) subsurface burial of labile organics in hadal sediment fueling intensified microbial activity. I understand that the hadal research is advancing rapidly, and that recent publications may not find their way into manuscripts that are being developed concurrently. But fortunately, the authors still have a chance to do that.

In conclusion, recommend the manuscript is accepted with relatively minor revision. Most importantly the authors should place their study in the context of other recent and highly relevant publications.

Response: Thank you for the positive feedback. Below we have addressed your detailed comments and specified several places where we added relevant papers in the revised manuscript.

More specific comments are listed below.

Abstract:

Line 79; I suggest changing “most mysterious” to “unique”.

Response: This is changed in the revised manuscript.

Line 86-89 & 90-93; I acknowledge the novel contribution of this high-quality study (See above/below). But conceptually, the fact that earthquake-triggered turbidity in hadal trenches result in subsurface layers of labile organic material and intensified microbial activity – and intensified burial and subduction of organic material has been documented and discussed in recent publications. For example;

*Zabel et al. (2022) High carbon mineralization rates in seafloor Hadal sediments – result of frequent mass wasting. *Geochem, Geophy, Geosystem* 23. Where some of the evidence originates from gravity cores that have been recovered from the Japan trench and include the SMTZ. I am particular surprised that this paper ins’ t considered in this submission.*

*Oguri et al (2022) Sediment accumulation and carbon burial in four hadal trench systems. *JGR Biogeosciences* 127. The manuscript, target mores sites along different trench systems, it is estimated that carbon burial rates in hadal trenches are 70 times higher than the global average for the deep ocean.*

*Schauberger et al (2021) Spatial variability of prokaryotic and viral abundance in the Kermadec and Atacama trench regions. *Limnol Oceanogr* 66: 2095-2109.*

None of these publications are acknowledged or cited in the current manuscript., but I find that the current submission should place their findings in the context of these recent advances.

Response: Thank you for your suggestions. We specifically value the work by Zabel et al. (2022) and Oguri et al. (2022). The three papers are now cited in the revised manuscript.

Oguri et al. (2022) as ref. ⁴ at Lines 85, 106 and 166;

Zabel et al. (2022) as ref. ¹⁴ at Lines 108, 166 and 206;

Schauberger et al (2021) as ref. ³³ at Line 185.

Line 154-159; This point has been presented and discussed in two recent papers of Zabel et al 2022 & Oguri et al 2022. Referencing and relating to these studies would substantiate your claims.

Response: Thank you. These two references are added at several relevant places in the manuscript.

Line 194-202; Can these claims be substantiated (no references are included, but they are presented as facts), or is these hypotheses? Please be specific.

Response: These texts are added with references at where we find relevant at Lines 198-210, where we attempted to hold the relevant researches together to provide a framework for our story, which is that earthquake-triggered turbidites enhance methanogenesis via various chemical (dissolved carbon supplement) and physical (sediment compaction) mechanisms.

Line 211-213; Evidence in the form of sequencing data (barcoding or metagenomics) or specific biomarkers for methanogens would have strengthened the proposed concept significantly. I don't know if there are any initiatives pursuing this for the recovered samples

Response: This topic will be pursued by our colleagues from IODP Exp. 386.

Line 220; I suggest changing “atmosphere” to “bottom water”. I acknowledge that “quickly” may mean different things for different people. But “quickly” is unspecific and venting of deep oceanic waters is on time scales of centuries to millennia.

Response: This suggestion is appreciated. This is changed in the revised manuscript at Lines 227-230:

*“The earthquake-triggered process cascade prolongs the residence time of labile carbon¹⁴, which would have been otherwise consumed and released into the **bottom water on time scales of centuries to millennia**, and thereby promotes carbon aging and subduction in the trench sediments (Fig. 4a).”*

Line 227-231; Are there any hardcore evidence that authentic carbonate precipitation is occurring in hadal sediments? Or is this another hypothesis/claim? Please be specific.

Response: Yes, we have added photos of the authigenic carbonate precipitation in the Japan Trench sediments as Supplementary Fig. 5:

Supplementary Fig. 5. The observed authigenic carbonates (ikaites) in Sites M0084 (a & b), M0088 (c & d), and M0087 (e).

Line 236; A reference here would be appropriate

Response: We have now referenced (Plank & Manning, 2019) and (Debret et al., 2022) at Lines 247-248, where the seawater-derived, OC-containing fluids in the subduction zones are discussed.

Line 254-256; that is not correct – there have been more “coordinated international research efforts that perform high spatiotemporal resolution investigations and sampling of a hadal oceanic trench” For instance, the cruises SO261 and TANI711 –that explored the deposition dynamic, carbon burial rates and carbon mineralization rates as well as microbial communities structures in sediment along the axes of the Atacama Trench and the Kermadec trench and relating condition to adjacent abyssal refences sites – a long list of papers from these expeditions have been published and more are in the pipeline. I fully acknowledge that this was not the impressively long cores that you obtain with the IODP efforts but “only” gravity cores, multiples cores and box cores. But the spatiotemporal variability along the trench axes and linkage to mass wasting were a core focus of these expeditions.

I suggest eliminating this sentence.

Response: Thank you. This sentence is removed in the revised version.

Comments from Reviewer #2

Review for Bao

I really appreciate to read a follow up study of the paper by Bao et al. On Hadal zone from a few years ago. It gives a sense of active progress on a key issue. That of turbidities and their role in transport (most?) organic carbon to trench.

The introduction is very clear, the goal is clearly stated.

Response: We appreciate your continuous attention for our research and your positive feedback.

I felt that the text is very interesting, but it reads sometimes like a review. You may spend a bit more time on your figures, explain very clearly what are your data, and do not mix so much your own work with that of your colleagues. This would bring a bit more clarity and focus to the main text.

Response: Thank you for your suggestion. We have now made minor modifications and added references where we find appropriate throughout the manuscript.

I was fascinated by the process-oriented study in hadal zone. The topic is very important, and the authors may be the only one focusing on this. The article reports a DOC compensation depth in the Japan hadal zone, they have evidences for carbon sink swaps (from organic carbon to carbonate) which is all very fundamental and important.

Response: Your positive feedback is appreciated.

The issue I see is that some may say "we already know carbon matures and produces a bit of methane". And I would agree that the way the text is written does not really answer this potential criticism, or at least does not make clear enough why such criticism does not affect the work. What you did here is not just show the OC transforms and produces CH₄. You should highlight the combination of tools, the focus on a very hard to study field site, that you have actually advanced our understanding of OC maturation in this very specific location. You should answer those points right away, be clear about your contribution.

Response: Thank you very much for helping us to clarify the focus of this manuscript. Several modifications were made regarding your suggestion:

- Highlighting the technical advancement of this study at Lines 264-268:
*"Our presented data and results, **acquired through advanced drilling and radiocarbon measurement techniques**, offer a comprehensive perspective of the hadal trench carbon cycle in the deepest and least explored environments on our planet, where geological events such as great subduction zone earthquakes may serve as key modulators."*
- Stating that our study shed light on a detailed process in the hadal trenches at Lines 268-271:
"The contribution from our newly-discovered carbon reservoirs in the Japan Trench sediments provide essential, detailed information regarding how intensive earthquake-induced SOC accumulation leads to enhanced microbial activities in the deep biosphere."
- Mentioning the significance of the DOC compensation depth and carbon sink swap occurring above the subduction zones at Lines 271-277:
"The occurrence of the dissolved carbon compensation depth indicates intensive carbon maturation and alteration in the trench subsurface sediments. These processes occur at the inlet to the subduction zone systems, where fresh materials are irregularly introduced by geological events. The cascade of the earthquake-triggered physical and chemical processes accelerates the transformation between carbon in different forms (i.e., 'carbon sink swap') at the shallowest setting of the subduction zone systems."
- Raising further questions and outlooks at Lines 277-281:
"Meanwhile, open questions remain regarding whether, and how much of, the carbon in the trench sediments is subducted in its various forms. We suggest that the earthquake-enhanced dissolved carbon reservoir in the trenches will constitute a potential source for the deep carbon reservoir in the lithosphere or mantle."

The conclusion is disappointing. It is full of "platitudes" and generalities unrelated to your work and not sexy at all (to me), they actually undermine your work greatly. What's sexy in your work is the novelties, the creative concepts blend with solid data, the amount of work you did. You should conclude your work with a big highlight on some of the key findings you made, and trust that we understand this is important. We do not need generalities to feel secure. Make us insecure instead, and show us that you are pushing forward in directions that no one is even looking at...

I think the study would benefit from a bit more work on the structure: your data first, and then

a bit more discussion. Here you tried to merge everything to an extreme degree and you lost clarity in my opinion, it dilutes your contribution.

I'd be happy to check a final revised version.

Best

Matthieu Galvez

Response: We delightfully accept your suggestion. The conclusion part is now substantially revised by putting more emphases on our data and discovery at Lines 264-281, with detailed explanations above.

"supporting the deep biosphere metabolites" does not really make sense. Metabolisms maybe.

Response: Thank you for the correction. Changed to metabolisms.

L207: "compensation depth of labile DOC" I find this notion intriguing.

Response: Thank you. We will do more works in the future regarding how these DOC are produced and utilized in the deep subsurface.

L215 – 222: very interesting!

Response: Thank you.

L227: do you mean authigenic?

Response: Yes. This is now corrected.

L227-229: for a "carbon sink swap" the other way around, check out Galvez et al 2012: carbonate to graphite.

Response: Thank you. We have stated that the carbon may be eventually transformed to graphite at Lines 239 and 255, where this paper is referenced as ref. ⁴⁷:

47. Galvez, M. E. et al. Micro- and nano-textural evidence of Ti(-Ca-Fe) mobility during fluid-rock interactions in carbonaceous lawsonite-bearing rocks from New Zealand. *Contrib. to Mineral. Petrol.* 164, 895–914 (2012).

L260-269: very disappointed by this: this sounds like this could have been written elsewhere. I felt throughout that your research strength is that it focuses on a the detailed processes happening in the hadal zone, the inlet to the subduction system. Even though everything you mention is correct, I would not conclude this work by those generalities. Conclude by stressing again your contribution. Mention again the notion of a DOC compensation depth, the notion that carbon sink swap may exist as shallow as on the seafloor and pore water. Those notions and the data that you provide to support them are truly innovative, finish with that.

Response: Thank you for the encouragement. We have now revised the conclusion part by putting more emphases on our contribution at Lines 264-281:

“Our presented data and results, acquired through advanced drilling and radiocarbon measurement techniques, offer a comprehensive perspective of the hadal trench carbon cycle in the deepest and least explored environments on our planet, where geological events such as great subduction zone earthquakes may serve as key modulators. The contribution from our newly-discovered carbon reservoirs in the Japan Trench sediments provide

essential, detailed information regarding how intensive earthquake-induced SOC accumulation leads to enhanced microbial activities in the deep biosphere. The occurrence of the dissolved carbon compensation depth indicates intensive carbon maturation and alteration in the trench subsurface sediments. These processes occur at the inlet to the subduction zone systems, where fresh materials are irregularly introduced by geological events. The cascade of the earthquake-triggered physical and chemical processes accelerates the transformation between carbon in different forms (i.e., 'carbon sink swap') at the shallowest setting of the subduction zone systems. Meanwhile, open questions remain regarding whether, and how much of, the carbon in the trench sediments is subducted in its various forms. We suggest that the earthquake-enhanced dissolved carbon reservoir in the trenches will constitute a potential source for the deep carbon reservoir in the lithosphere or mantle."

REVIEWERS' COMMENTS

Reviewer #1 (Remarks to the Author):

The authors have satisfyingly address all my concerns, I encourage acceptance of the manuscript for publication

Reviewer #2 (Remarks to the Author):

Dear authors and editors,

I have checked your revision after my previous review. I felt my comments and suggestions have been considered. I will be pleased to see this paper published as it brings valuable data to the community, and it will also stimulate the community toward a direction of research which remains underrepresented (oceanographic approaches to the subduction carbon cycle, the inlet of the system).

Response to Reviewers

We appreciate the feedbacks from the Reviewers concerning our manuscript “*Earthquake-enhanced dissolved carbon cycles in ultra-deep ocean sediments*”. The responses to the Reviewers’ comments are listed below, with comments listed first in *blue italics*, followed by our response in black. We are pleased that both Reviewers are satisfied with our revised manuscript.

Comments from *Reviewer #1*

The authors have satisfyingly address all my concerns, I encourage acceptance of the manuscript for publication

Response: We thank you again for your efforts reviewing our manuscript.

Comments from *Reviewer #2*

Dear authors and editors,

I have checked your revision after my previous review. I felt my comments and suggestions have been considered. I will be pleased to see this paper published as it brings valuable data to the community, and it will also stimulate the community toward a direction of research which remains underrepresented (oceanographic approaches to the subduction carbon cycle, the inlet of the system).

Best

Matthieu

Response: Thank you. Your comments have helped us to substantially improve our manuscript.